# Telomere Length Abnormality: Investigating Approaches and Correlations with Cancer, Bone Marrow Failure and Hematological Malignancies

**DOI:** 10.3390/biomedicines13123009

**Published:** 2025-12-08

**Authors:** Corrado Tarella, Dario Ferrero, Maria Beatriz Herrera Sanchez, Alessia Rita Canestrale, Sharad Kholia, Lorenzo Silengo, Enrico Derenzini, Irene Ricca

**Affiliations:** 1Oncohematology Division, IEO European Institute of Oncology IRCCS, 20141 Milan, Italy; 2Telomere Research Unit at Molecular Biotechnology Center, 10126 Turin, Italy; 3Department of Molecular Biotechnology and Health Sciences, University of Torino, 10126 Turin, Italy; 4Molecular Biotechnology Center, University of Torino, 10126 Turin, Italy; 52i3T, Società per la Gestione dell’incubatore di Imprese e per il Trasferimento Tecnologico, University of Torino, 10126 Turin, Italy; 6Department of Health Sciences, University of Milan, 20122 Milan, Italy; 7Expert Center for Congenital Hemorrhagic Disease (CHD) in Pediatrics and Pregnant Women—Transfusion Medicine, University Hospital of the City of Health and Science of Turin, 10126 Turin, Italy

**Keywords:** telomere, telomerase, telomere length measurement, non-hematologic cancers, bone marrow failure, hematological malignancies

## Abstract

Proper telomere maintenance is crucial for ensuring healthy cellular function. Telomeres have a tendency to reduce in length with cellular aging. Moreover several factors may promote telomere attrition. Other conditions, primarily due to genetic and inherited origins, can be characterized by unusually long telomeres. Both shortening and elongation of telomere length (TL) may lead to increased risk of cancer occurrence or cancer progression. Additionally, some hematopoietic dysfunctions may also be associated with telomere abnormalities. This review is aimed to describe and discuss main aspects of TL, in relation to carcinogenesis. The initial section describes main current methods for TL assessment, since the accurate and reliable TL measurements is a crucial issue in TL research. The various studies describing the association between TL and cancer risk are then reported and critically illustrated, with special interest on TL shortening in hematological malignancies, as well as in some peculiar non-malignant dysfunctions. Hence, a systematic analysis of the broad contribution of TL to cancer development is extensively appraised.

## 1. Introduction to Telomere and Telomerase: Structure, Functions and Relevance for Cancer

Telomeres are specialized nucleoprotein structures located at the termini of eukaryotic chromosomes, playing a crucial role in maintaining chromosomal DNA integrity [1]. Telomeric DNA is a segment of long tandem repeated, guanine-rich DNA sequences combined with proteins. This complex caps the ends of chromosomes, and its primary function is to protects chromosomes from damage. The length of telomeres varies among chromosomes and between individuals. However, the telomeric repeat DNA sequence is highly conserved across all eukaryotes [2,3,4]. This conservation indicates that telomeres are ancient and evolutionarily preserved structures of great importance, primarily ensuring genome integrity [5]. Furthermore, emerging evidence suggests that their functions extend beyond the maintenance of chromosome stability, encompassing regulatory roles in stress-responsive signaling pathways and gene expression modulation [6,7,8].

Proper telomere maintenance is crucial for ensuring healthy cellular function. In spite of this, telomeres tend to reduce in length. This happens through two main distinctive mechanisms [9]. The first one ensues because DNA polymerase is unable to fully replicate the 3′ end of the DNA strand, and consequently telomeres shorten with cell division. Thus, telomeres physiologically shorten with cell proliferation, with an estimated loss of approximately 30–150 bp with each cell division [10,11,12]. Due to this trait, telomere length (TL) is regarded as a biomarker of cellular aging [13]. The second main way of TL shortening is represented by the activity of several factors known to influence TL, in particular oxidative stress, inflammation, carcinogens, and cytotoxic drugs [14,15,16,17,18,19,20,21,22,23,24,25,26]. Moreover, lifestyle factors, including obesity and weight loss, cigarette smoking, lack of sleep, depression, as well as dietary pollution and environmental, occupational, and health risk factors can cause human telomeres to shorten prematurely [25,27,28,29,30,31]. Indeed, TL is an indicator of biological age further to chronological age, as depicted in Figure 1. In this context, it should be recalled that telomere erosion is more pronounced in men compared with women and females have, on average, longer telomeres than males. This sex differences is paired with differences in the average life expectancy between the sexes [32,33,34,35].

The enzyme telomerase, among its various pleiotropic functions, maintains telomere length (TL) by extending the 3′ ends of chromosomal telomeric DNA. Telomere sequences are restored through the action of telomerase reverse transcriptase (TERT) and its integral template-containing telomerase RNA (TER) which provides the template for extending the ends of chromosomes [37,38,39]. Thus, telomerase is crucial for preventing telomere shortening. However, human telomerase (*hTERT*) activity varies depending on the cell type. For instance, in embryonic cells, telomerase remains active, ensuring consistent maintenance of TL [40,41,42,43]. In adult stem cells, its activity is limited, thereby only partially compensating for telomere shortening (TS) [44]. In contrast, *hTERT* is usually inactive in somatic cells, leading to progressive telomere shortening that exceeds elongation at both cellular and tissue levels [45,46,47]. Once telomeres reach a critical length, they become dysfunctional, triggering a DNA damage response that induces senescence, characterized by reduced cell proliferative capacity and the development of a senescence-associated secretory phenotype (SASP) [13,48,49]. At this stage, cells undergo cell cycle arrest or apoptosis [48,50,51,52,53]. However, telomere dysfunction and the resulting genomic instability may, in some incipient cancer cells, promote the acquisition of genetic mutations that enable the bypass of cell cycle checkpoints. In these cells, telomerase activation is essential for TL restoration and the recovery of proliferative capacity [54,55]. Telomerase-positive cells can thus escape telomere crisis and continue proliferating as transformed cells with genomic rearrangements that ultimately give rise to malignant cancer cells.

Based on the above concepts, it clearly comes out that telomeres are strongly associated with cancer development. In particular, even if TL shortening may act as a protective mechanism inducing dysfunctional cells toward cell cycle arrest and apoptosis, on the other hand, short telomeres bring about increased genomic instability that may ultimately cause carcinogenesis [52,56,57]. Inversely, cells with long TL have high proliferative potential, and may extensively grow and acquire mutations suitable for cancer development. Thus, both shortening and elongation of TL lead to increased cancer risk. Considering this, it is indicative that both short and long TL have been reportedly associated with a high risk of cancer occurrence. Hence, information on TL and telomere maintenance mechanisms and detailed investigation on factors potentially affecting TL are essential in order to improve our knowledge on the prevention of age-related diseases, including cancer. The role of TL in carcinogenesis and the relevance of TL investigations with reference to cancer occurrence and cancer development are outlined in Figure 2.

This review is aimed to describe and discuss the main aspects of TL and telomere function, in relation to cancer development. Since an unresolved issue in telomere research is the development of accurate and reliable TL measurements, the initial section of the review analyzes different aspects of current methods for TL assessment. The various studies reporting the association between TL and cancer risk are then reported and critically illustrated, with special interest on TL shortening in hematological disorders.

## 2. Main Techniques Used for TL Measurement and Analysis

Selecting an appropriate method for telomere length (TL) measurement requires balancing ease of use with the level of information obtained. Large-scale epidemiological studies benefit from high-throughput techniques capable of rapidly analyzing numerous samples. Conversely, in-depth investigations seeking to elucidate the mechanisms of telomere function might prioritize methods that provide more detailed information, particularly those assessing the distribution of telomere lengths across all chromosomes, with special attention to the shortest telomeres. A variety of techniques have been developed for the study of telomeres, each with distinct advantages and limitations. This review discusses the major TL measurement strategies, highlighting their technical requirements, reliability, and reproducibility [58,59,60,61]. Recent advances introducing novel methodologies are also considered alongside established approaches [62,63]. Table 1 summarizes the main techniques for TL assessments described herein.

### 2.1. Telomere Restriction Fragment Analysis by Southern Blotting

Telomere Restriction Fragment (TRF) analysis, based on the detection of the repetitive TTAGGGn telomeric sequence, uses a labeled probe to identify these regions following DNA digestion [2]. This method, often regarded as the “gold standard” for TL measurement, provides both average and absolute TL by analyzing the intensity of telomeric smears on Southern blots [11]. However, TRF analysis has several limitations. It struggles to detect critically short telomeres, those most relevant to cellular health (under 2 kb), due to constraints in probe hybridization [72,73]. Moreover, the method requires a substantial amount of DNA (at least 1 µg), involves a lengthy and labor-intensive procedure, and has low throughput, typically accommodating no more than 30 samples per run. Consequently, Southern blotting is impractical for large-scale studies [72]. Although commercial kits have been developed to improve reproducibility across laboratories, the choice of restriction enzymes can still significantly affect the results.

### 2.2. Quantitative Polymerase Chain Reaction

Quantitative Polymerase Chain Reaction (Q-PCR) offers a user-friendly and efficient approach for TL analysis, requiring only minimal amounts of starting DNA. It measures the ratio of the telomere signal (T) to that of a single-copy reference gene (S), producing a T/S ratio that reflects the average TL rather than the absolute length in kilobases [64,74]. The high-throughput nature of this technique makes it particularly well-suited for large-scale population studies. However, Q-PCR has several limitations. First, it provides relative, not absolute TL values; accurate length determination requires comparison with a reference cell line whose telomere length has been measured using another method [74]. Second, inter-laboratory variability can arise from the use of different reference genes, some of which may not be true single-copy genes, thereby influencing the T/S ratio. Third, Q-PCR does not capture information on critically short telomeres, which are highly relevant to cellular health. Finally, the reliability of Q-PCR decreases in cancer studies, as the reference gene itself may be affected by chromosomal abnormalities commonly observed in cancer cells [75]. Therefore, Q-PCR is best-suited for the analysis of normal diploid samples with stable karyotypes.

### 2.3. Fluorescence In Situ Hybridization

Quantitative Fluorescence In Situ Hybridization (Q-FISH) provides a comprehensive approach to TL measurement [65]. This technique uses fluorescently labeled probes to detect telomeric sequences on chromosomes. Interphase Q-FISH measures telomere fluorescence intensity in individual cells under a microscope, whereas metaphase Q-FISH offers higher accuracy by assessing TL at each chromosome end in dividing cells [76]. Commercially available adaptations, such as Telomere Analysis Technology (TAT), enable high-throughput analysis across various cell types and tissues [77]. By combining fluorescent probes with flow cytometry, a method known as Flow-FISH, it is possible to determine the average TL of large cell populations simultaneously [66,78,79]. Furthermore, by using distinctly labeled antibodies, it is possible to identify different cell subpopulations concurrently with TL evaluation [79]. However, FISH-based methods also have limitations. They are time-consuming and require specialized, expensive equipment. In addition, the probes may hybridize to nonfunctional telomeric-like sequences within chromosomes, potentially compromising measurement accuracy [80,81]. Moreover, precise optimization of flow cytometry parameters and stringent protocol control are essential to ensure reliable results. Despite these constraints, FISH techniques have played a pivotal role in telomere research, owing to their broad applicability and ability to provide detailed, cell-specific information [79,82,83].

### 2.4. Telomere Dysfunction-Induced Foci Analysis

Telomere dysfunction-induced foci (TIF) analysis is a valuable technique for investigating DNA damage at telomeres. Typically applied to cells or tissue sections, this method employs two antibodies: one targeting a shelterin protein, such as TRF2, which normally protects telomeres, and another recognizing DNA damage markers, such as **γ**H2AX or 53BP1. Optionally, a telomeric repeat probe may be used alongside the DNA damage marker [67]. The co-localization of these signals indicates that DNA damage is occurring at telomeres rather than randomly throughout the genome. An elevated number of co-localized foci suggests telomere dysfunction, which may serve as a biomarker for monitoring the efficacy of telomerase-inhibiting therapies in clinical trials. It is important to note, however, that TIF analysis does not measure telomere length. Instead, it identifies telomeres that are critically short or uncapped, displaying feature characteristics of damaged DNA. When combined with methods such as TRF analysis or Q-FISH, TIF analysis can provide complementary insights into telomere status, particularly relevant when telomeres are severely shortened.

### 2.5. Single Telomere Length Analysis

Single Telomere Length Analysis (STELA) stands out for its ability to measure the length of individual telomeres on specific chromosome ends [68]. Unlike other methods that provide only average values, STELA yields absolute TL and focuses particularly on critically short telomeres through a combination of molecular techniques. This approach has generated important insights, including the identification of extensive allelic variation in telomere length and the detection of ultrashort telomeres in aged human cells [68]. Another advantage is its ability to assess telomere length on specific chromosomes even when only limited sample material is available. However, STELA also presents several limitations. Designing highly specific primers for each chromosome end is technically challenging, restricting the number of telomeres that can be analyzed. To address this issue, the Universal STELA (U-STELA) method was developed in 2010, enabling the study of all telomeres simultaneously [84]. Although U-STELA provides broader coverage, it struggles to accurately measure telomeres longer than 8 kb [74]. Furthermore, both STELA and U-STELA require manual analysis of hundreds of telomere lengths, making the process labor-intensive and time-consuming. To overcome this limitation, the High-Throughput STELA (HT-STELA) method was recently developed [62,69,85]. This technique employs a PCR-based assay to amplify telomeric regions and a bioanalyzer to automatically assess TL. Despite offering increased automation and efficiency, HT-STELA remains costly and is restricted to targeted chromosomes for TL measurement.

### 2.6. Telomere Shortest Length Assay

Telomere Shortest Length Assay (TeSLA) represents a significant advancement in telomere length measurement, overcoming limitations of other methods by offering heightened sensitivity and accuracy. TeSLA can detect a broader range of telomere lengths, spanning from less than 1 kb to 18 kb, inclusive of critically short telomeres, without interference of internal telomere sequences (ITSs) [69,85]. Moreover, it provides a more precise depiction of telomere distribution. Requiring minimal starting DNA (less than 1 µg), TeSLA employs a blend of enhanced ligation, digestion, and Southern blotting techniques coupled with a highly sensitive probe [69]. Furthermore, user-friendly software automates telomere length measurement, empowering researchers to track subtle changes in telomere dynamics over time [85]. TeSLA has demonstrated success across various studies, including the monitoring of telomere changes during human aging in blood cells (PBMCs), investigation of telomere dynamics in colon cancer progression, analysis of telomeres in telomere-related diseases such as idiopathic pulmonary fibrosis, and exploration of telomeres in TERT knockout mice and other organisms [85]. Although TeSLA, similarly to STELA and U-STELA, operates at a lower throughput, it yields invaluable insights, particularly in human and most animal studies. Notably, it excels at capturing changes in the shortest telomeres, which are pivotal for comprehending disease development and potentially facilitating earlier interventions. Nevertheless, this approach has additionally some drawbacks: it is a time-consuming and labor-intensive assay and it fails to detect alternative lengthening of telomeres (ALT), adopted by some cancer cells for telomere maintenance.

### 2.7. Single-Cell Telomere Length Measurement

Single-cell telomere length measurement with pqPCR (SCT-pqPCR) is a technique developed for the measurement of TL in a single cell, independent of its ability to divide. Since the amount of DNA in a single cell is minimal (approximately 10 pg), a pre-PCR amplification step is performed to increase the quantity of DNA available for analysis, followed by a qPCR. The relative TL is expressed as the ratio between the telomere signal and that of a reference gene (such as *Alu* for human cells or *B1* for mouse cells) [70,86]. Despite its potential, this innovative technique is limited by the high level of technical expertise required.

### 2.8. Optical Mapping

This technology enables the direct visualization of DNA structure and sequence [71]. DNA molecules are aligned in nanometer-scale channels, labeled with fluorescent dyes, and observed using high-resolution microscopy. The Cas9n enzyme, a variant of the Cas9 nuclease, is employed to measure TL. It cleaves one strand of the telomeric DNA, leaving the complementary strand intact, while a specifically designed fluorescently labeled probe targeting the sequence adjacent to the cleavage site is introduced [71]. This probe binds to the cleaved DNA, marking the telomeric region with fluorescence. Following labeling, the telomeric DNA is elongated and fixed within the nano-channels, allowing high-resolution visualization and analysis. Techniques such as fluorescence microscopy or other advanced imaging methods are then used to observe the labeled telomeres and accurately measure their length [87,88]. However, the operation of nano-channel systems is highly complex, and DNA cleavage patterns may be uneven due to variations in the targeting efficiency or specificity of Cas9n. As a result, this procedure is best-suited for highly specialized research centers and selected studies.

## 3. Short and Long Telomeres: Pathogenetic and Prognostic Role in Non-Hematologic Cancers and Hematological Disorders

### 3.1. General Concepts

Telomeres are DNA structures essential for the maintenance of healthy cellular functions and prevention of age-related diseases, including cancer. For its critical role, telomere and telomere length (TL) have been extensively studied for their possible association with cancer occurrence. The fate of telomere in somatic cells is to gradually shorten with cell division leading to senescence and ultimately to cell death. This is considered a protective mechanism against tumorigenesis. Indeed, senescent and vulnerable cells are driven to apoptosis and death by p53 activation, which is induced by the DNA damage response when TL shortening comes to critical values [9,52,89,90,91]. However, short telomeres are also known to increase genomic instability, and this may instead promote carcinogenesis [92,93,94,95]. Mounting telomere dysfunction with chromosomal instability may provoke the acquisition of genetic mutations in p53 or other checkpoint proteins allowing the cells to overcome senescence and continue to proliferate [37,96,97]. The stable escape from the apoptotic path can be then sustained by the anomalous transcriptional activation of telomerase. In few other situations, the activation of the alternative lengthening of telomeres (ALT) may take place [54,55,98,99,100,101]. The latter is the so-called telomerase-independent telomere maintenance mechanism utilizing the DNA homologous recombination repair pathway. Whatever the mechanism, telomere function can be aberrantly reconstituted with restoration of the missing proliferative capacity. The outcome of these events is a telomerase or ALT-positive, transformed cell with a heavily rearranged genome having potentially tumorigenic genetic mutations. Thus, TL shortening with its pro-apoptotic effect may basically act against cancer promotion, while its associated genetic instability may favor over time an increased risk of tumor growth and progression.

On the other hand, cells with longer telomeres have higher proliferative potential and are more prone to acquiring mutations, due to the prolonged division before entering senescence [102,103]. Supporting evidence to this concept arises from recent studies showing that individuals with genetically longer telomeres might face a heightened risk of certain cancers [104,105,106,107]. Thus, the relationship between telomere length and cancer risk in humans is intricate and both shortening and lengthening of telomeres could increase cancer risk, as already recapitulated in Figure 2. This may at least in part explain some discrepant results that have been reported so far. Nevertheless, the causal role of telomere length, either short or long, in the occurrence of cancer is now well-established. Besides this, there is no conclusive data on whether telomeres are lengthened or shortened in cancer patients and in tumor cells. Again, this is in part due to the distinct tumorigenic role of both short and long telomeres [108]. In addition, there might be basic differences in TL among tissues and even among cells of the same tissue or of the same tumor [109,110,111,112,113]. The marked variations among cancers and within cancer subtypes preclude general assumptions on the correlation between TL and cancer development and/or prognosis. Moreover, TL assessment can be understandably influenced by the type of biological material investigated [114]. Most studies have been performed on peripheral blood (PB) cells, either unfractionated mononuclear cells or on selected lymphocytes, based on the assumption that TL of PB cells reflects TL of the whole organism and of cancer cells as well. However, direct evaluation of TL on neoplastic tissue and on adjacent normal tissue may offer quite distinct results. Lastly, various methodologies for TL evaluation have been developed over the last decades, as detailed in the previous section. Thus, the strategies for TL measurement should be carefully considered when comparing the results on TL and cancer reported in different studies [114]. All these aspects together explain the scarcity of clear-cut indications on both the pathogenetic and the prognostic value of TL changes in cancer. The relevance of these matters and the lack of conclusive understandings explain the increasing number of reports that have been produced over the last decade on the association of both shortening and lengthening of telomeres with tumor development and progression. The most relevant studies dealing on this issue are reported here.

### 3.2. Telomere Length in Main Non-Hematologic Cancers

TL of PB leukocytes is accepted to be highly indicative of TL of the other tissues and organs. Thus, several studies have employed PB cells as useful cell sources to investigate the correlation between TL and risk of cancer, including non-hematologic cancers [115]. Overall, both retrospective and prospective surveys have confirmed the association of TL and non-hematologic cancer occurrence, although apparently conflicting results have been reported. For a few cancers, a clear association with long telomere in PB cells has been documented [115,116,117,118]. Moreover, long TL has been reported in PB lymphocytes in distinct non-hematologic cancers, suggesting that long TL in lymphocytes may have implications in terms of increased susceptibility and poor prognosis [119,120]. A recent meta-analysis has documented that long leukocyte TL is associated with an increased risk of all types of lung cancer, with a particular increase for adenocarcinomas and lung cancer in never-smoker subjects [121]. This observation takes further support by a recent report showing that occupational exposure to certain pesticides, namely lindane and the insecticide diazinon, is associated with increased leukocyte TL, although other types of insecticide or herbicide may induce leukocyte telomere shortening [122]. Notably, diazinon exposure might be associated with increased risks of lung malignancies. This suggests possible correlations to exposure to some pesticides, longer leukocyte TL, and lung cancer.

Recently, studies have been addressed with mounting interest to the genes essential for telomere maintenance. The chromosome 5p15.33 region is of particular attention. It includes the *h-TERT* gene. Multiple germline variants have been identified by genome-wide association studies (GWASs) within this region and found to be associated with cancer predisposition or protection [123,124,125,126]. These surveys have been performed on large-scale analysis on various cancer types, mostly of non-hematologic origin. A more recent report has thoroughly characterized variants within *h-TERT* introns [127]. By regulating *h-TERT* splicing, these variants may contribute to modulating cell longevity and replicative potential. In fact, genetically determined TERT functions can be differently exerted according to the context of tissue-specific conditions. Thus, the study uncovers the complex regulation of TERT functions and the contribution of *h-TERT* germline variants to cancer susceptibility and to telomere biology as well. Important insights on genetic regulation of TL have been obtained by a wide study combining TL evaluation by quantitative PCR and whole-genome sequencing measurements from 462,666 UK Biobank participants [128]. The study identified 64 variants and 30 genes significantly associated with TL. Notably, 16% of these genes are known drivers of clonal hematopoiesis, with somatic variants associated with either lengthened or shortened telomeres. The results further emphasize the impact of rare variants on telomere length. Thus, a few germline pathogenic variants have been identified in genes essential for telomere length maintenance and function. The biological consequences of these variants range from short or dysfunctional telomeres with reduced cellular replicative potential to long telomeres and increased cellular replicative capacity and these biological abnormalities may cause a broad spectrum of clinical disorders, including cancers.

In order to clarify these genetically predetermined disorders, two broad classes of human disease have been proposed by Savage and Bertuch on behalf of Team Telomere and the Clinical Care Consortium for Telomere-Associated Ailments (CCCTAA) [129]. According to these Authors, the first group is represented by the well-known telomere biology disorders (TBDs). This group includes the conditions that manifest in individuals with abnormally short or otherwise dysfunctional telomeres and are associated with a spectrum of life-threatening conditions, including bone marrow failure, liver and lung disease, as well as cancer (see next chapter). The second main group, identified with the term “cancer predisposition with long telomeres (CPLT)” recognizes the condition caused by germline variants in telomere biology genes, associated with longer-than-average functional telomeres. These disorders are associated with elevated risk of a variety of cancers, primarily melanoma, thyroid cancer, sarcoma, glioma and several lymphoproliferative neoplasms as a consequence of excessive telomere elongation and increased cellular replicative capacity.

The Mendelian randomization (MR) approach has been used in several studies to investigate genetically determined telomere variations and the risk of cancer. A large MR analysis on 420,081 cases and 1,093,105 controls revealed an association between genetically predicted longer telomeres and the risk of eight distinct cancer types tested, with the strongest association observed for glioma (OR 5.27), serous low-malignant-potential ovarian cancer (OR 4.35), lung adenocarcinoma (OR 3.19), and neuroblastoma (OR 2.98) [130]. Overall, associations were stronger for rare cancers and at tissue sites with a low rate of stem cell replicative potential. The notion that longer telomere length due to germline genetic variants is associated with a higher risk of site-specific cancers has been confirmed by others. In particular, a recent two-sample Mendelian randomization study has found strong associations with thyroid (OR 2.49) and lung (OR 2.19) cancers [131]. To sum up, it is likely that longer telomeres increase risk for several cancers and this predisposition can be determined by germline genetic variants. On the other hand, other genetic variants may be responsible for abnormally short telomeres and this can be associated with a spectrum of non-neoplastic life-threatening conditions, and occasionally with some cancers as well.

Besides investigations on genetically determined telomere variations, many studies have been focused on the evaluation of TL in a variety of patients with cancer. In a fundamental study by Barthel et al., TL was analyzed in 18,430 samples from 31 types of cancer, mostly non-hematologic cancers, using data from The Cancer Genome Atlas (TCGA). Overall, 70% of these cancers had shorter telomeres compared to normal samples, and this was associated with increased telomerase activity [54]. In the remaining 30% of cases, telomere length was observed to be either the same or longer than average [54]. Several other studies have documented that short TL, mostly in blood cells but also in tumor tissue, is associated with increased incidence and/or poor outcome for several cancers, as summarized in quite a few meta-analysis [132,133,134,135,136].

Studies on tumor tissues have been remarkably convincing on the role of short TL in tumor progression. Particularly interesting is the analysis performed in repair-deficient colorectal cancers arising in Lynch syndrome (LS-CRC) and, for comparison, in microsatellite stable sporadic colorectal cancer (MSS s-CRC) and in benign colon mucosa, as well [137]. A progressive decrease in mean telomere length in all cancer subtypes compared with adjacent normal mucosa was observed, with higher telomere shortening rates in LS-CRC compared to s-CRC. A similar study found that TL is shorter in colon cancer tissue than in the adjacent mucosa [138]. Notably, patients with a small TL ratio between tumor tissues and the adjacent mucosa were associated with increased overall survival. The results provide strong support to the view that telomere attrition is an early event in tumorigenesis for most gastro-intestinal tumors, particularly in CRC. Moreover, the degree of telomere attrition may also have prognostic implications.

Many studies have focused the attention on the role of telomere shortening in the initial steps of tumorigenesis. The study of TL in precancerous pancreatic lesions is particularly relevant [139]. The study shows that telomere shortening occurs in the early stages of pancreatic carcinogenesis and progresses with precancerous development. Further analysis with direct TL measurement on tissue samples has clearly demonstrated that the detection of short telomeres is one of the earliest genetic events in pancreatic carcinogenesis [139,140,141]. In several other studies, TL has been evaluated in leukocytes (LTL) of patients with pancreatic ductal adenocarcinoma (PDAC), the most common form of pancreatic cancer. Additional reports have shown an association between short LTL and increased PDAC risk, although long LTL has also been documented. At present, short TL is more convincingly associated with PDAC risk and indeed TL shortening might reflect lifestyle exposure to pro-cancerogenic substances or conditions [142]. A recent overview summarizes our main notions on the role of telomere and telomerase in various pre-neoplastic lesions [143]. Abnormal telomere length, with either shorter or longer telomeres, is found in pre-invasive lesions and appears to predispose one to cancer development. In particular, short TL has been reported not only in GI and pancreatic precancerous lesions but also in early lung, breast, and prostate cancers, whereas increased TL and telomerase activation are significant risk factors for glioma and glioblastoma development. Briefly, several studies have documented that pre-neoplastic lesions across different tissues are associated with the increased expression of TERT activity, abnormal TL and the occurrence of *hTERT* mutations, showing the critical involvement of telomeres and telomerase reactivation during the evolution from pre-neoplastic lesion to overt cancer.

The prognostic value of TL variations has been extensively investigated in breast cancer. The various results are summarized in a meta-analysis including nine reports involving a total of 3145 breast cancer patients, with TL assessed either in blood or in tumor tissues [144]. Data from this systematic analysis show that a significantly increased recurrence risk was observed among patients displaying shortening TL, along with a trend toward reduced overall survival (OS). Even more extensive is the literature on TL and telomerase reactivation in gastro-intestinal cancers [145]. TL has been found frequently shortened in GI tract cancers, especially in gastric and colon cancers although some controversial results have been reported. A clear association has been found for telomere length in the blood of Barrett’s esophagus patients and their likelihood of developing esophageal adenocarcinoma [146]. In addition, telomerase activity has been shown to be linked to tumor aggressiveness and chemotherapy response in esophageal cancer [147]. Similarly, TA was found to be significantly more common in gastric cancer and colon–rectal cancer compared to normal gastric mucosa, gastric polyps and colon–rectal polyps [148,149]. High telomerase activity along with short telomeres were associated with adverse prognosis in gastric cancer [150,151]. Several studies have evaluated TL in colon cancer tissues compared to adjacent normal mucosa and most primary tumors showed shorter telomeres compared to noncancerous tissue [152,153]. A moderate to high telomerase activity was also observed in colon–rectal tumors compared to normal colon tissues [154]. Again, both telomere shortening and high telomerase activity were found to be of prognostic value [152,154,155,156].

The abundant reports herein mentioned and discussed prove that telomeric length abnormalities are a hallmark of human non-hematologic cancer and its precursor lesions. Overall, TL shortening is more frequently encountered compared to lengthened TL in solid cancers. In any case, either too short or too long TL may influence cancer risk, depending on the type of cancer [115,117,135,136]. Main studies dealing with long or short TL in non-hematologic cancers herein presented are summarized in Table 2.

### 3.3. Telomere Length in Bone Marrow Failure

#### 3.3.1. Inherited Telomeropathies

Telomere and telomerase have been extensively investigated in the hematopoietic system. This has led to substantial improvements in our understanding of the basics of telomere biology as well as of how telomere function and telomerase activities are involved in development of bone marrow failure and blood malignancies. Accordingly, the recognition of a spectrum of telomere biology disorders (TBDs) or telomeropathies has been a major step forward. TBDs include a broad variety of infrequent disorders that result from inherited defects in the telomerase maintenance mechanism or the DNA damage response (DDR) system [157,158,159]. These germline-inherited conditions are characterized by accelerated telomere shortening, predisposition to organ failure syndromes, along with increased risk of neoplasms, especially hematological malignancies. Among TBDs, probably the most representative disorder is Dyskeratosis Congenita (DC), a genetically heterogeneous inherited disease. It was initially identified by mutations in dyskerin (*DKC1*), a pseudo-uridine synthase enzyme essential for the activity of telomerase and its RNA component, although mutations in several other genes resulting in reduced telomerase activity and short telomeres have been discovered as genetic etiology of DC [160,161,162,163]. Additionally, the clinical presentation and onset age are quite heterogeneous in DC and related TBDs and the common thread shared by these inherited disorders remains the germline mutations resulting in abnormal telomere biology. Indeed, it has to be underlined that the severity of symptoms in DC/TBDs is directly shaped by the severity of telomere abnormality. Thus, the careful study of clinical features associated with telomeropathies has offered additional information on the wide and crucial role of telomere dysfunction in several human diseases [163].

A common manifestation of TBD is bone marrow failure. Again, the severity is quite variable, ranging from one single lineage of cytopenia up to severe pancytopenia with possible malignant transformation into myelodysplasia (MDS) or even acute myeloid leukemia (AML) [164,165,166]. An additional hematological feature of short telomeres is the generation of clonal hematopoiesis (CH), indicating one more way of cancer promotion by short telomere. CH is characterized by clonal expansion of cell subpopulations, usually in the absence of cytopenias and dysplastic hematopoiesis and thus of indeterminate potential (CHIP) [167]. The clonal population shows somatic mutations in the same genes associated with hematological malignancies, namely MDS and AML. These variants occurring in normal subjects are more frequently observed in advanced age, whereas TBD-associated CHIP may occur at younger age. Subjects with CHIP are at an increased risk of developing hematological malignancies as well as coronary heart disease and other cardiovascular events, including stroke [168]. In summary, TBDs are cancer-predisposing multi-systemic diseases showing a high risk of progression into myeloid neoplasms [169].

In patients with TBDs, short telomere is the key hallmark, leading to premature cell senescence along with high risk of bone marrow failure and risks of myeloid neoplasms as well. Thus, TBDs might benefit from treatments aimed at lengthening stem cell telomeres. In this view, the recent report of a novel therapeutic approach for TBDs with immature CD34+ve cells treated with *ZSCAN4* is of great interest [170]. The protein ZSCAN4 is encoded by a mammalian-specific gene transiently expressed in embryonic cells, with a critical role in telomere maintenance, independently from telomerase, and embryonic cell stability with normal karyotype preservation [171]. Some observations indicate the possible activity of ZSCAN4 in adult cells as well [172,173]. The recently published report is a very preliminary observation in two patients with TBDs treated by infecting their mobilized CD34+ve cells with the temperature-sensitive Sendai virus (SeV) vector, encoding the human *ZSCAN4* gene. Telomere lengthening was observed in CD34+ cells following ex vivo *ZSCAN4* exposure. Moreover, longer telomeres in peripheral blood cells were detected following infusion of autologous ZSCAN4-treated CD34+ve cells [170]. This observation, even if at a preliminary stage, demonstrates that telomere lengthening is feasible and indicates that exposure of autologous CD34+ cells to *ZSCAN4* is a potential therapeutic intervention for TBDs and possibly other forms of telomere-associated hematopoietic disorders.

Besides DC/TBDs, there are other inherited disorders characterized by BM failure (IBNF) that have been well-known for a long time, namely Fanconi Anemia (FA), Scwachman–Diamond syndrome (SDS), and Diamond–Blackfan anemia (DBA) [174,175]. Despite the heterogeneity in the underlying genetic disorders, all inherited BM failures have a higher risk of MDS and/or AML development, with DC and FA showing the highest risk. Moreover, similarly to DC patients, TL is extremely short in subjects with non-DC IBMFs, such as FA and SDS, as well as DBA, the latter with TL not as short as in DC [176,177,178,179]. Thus, several inherited hematological disorders characterized by early TL shortening are associated with increased risk of developing blood cancer. In previous parts of this section, it has been mentioned that increased TL due to germline genetic variation may correlate with increased risk of malignancy, including the risk of CHIP development, similarly to what was observed in inherited telomeropathies [104,105,106,107]. An increased risk of familial cancer has been clearly documented in an autosomal dominant family with cutaneous malignant melanoma (CMM) carrying a pathogenic variant in the *h-TERT* promoter [180]. Other studies have observed strong association with some rare cancers and tissue sites characterized by low rates of stem cell division, such as glioma, serous low-malignant-potential ovarian cancer, lung adenocarcinoma, neuroblastoma, bladder cancer, and melanoma [130,163]. Taken together, the results from studies in inherited telomere disorders confirm that telomere dysfunction with either long or short TL is definitely prone to cancer development. In hematology, this risk seems particularly elevated in inherited BM failure disorders characterized by markedly short TL.

BM failure along with the risk of blood cancer development are the well-known clinical manifestations of TBDs. However, inherited short telomeres may also be associated with extra-hematological disorders, that need to be mentioned and described. For instance, DC, besides BM failure, is characterized by skin pigmentation, oral leucoplakia, nail dystrophy, gastro-intestinal hemorrhage, and increased risk of interstitial lung disease, cirrhotic liver disease, and solid cancers [181]. Other less-known syndromes are mainly characterized by defects at birth, although also BM failure is included among clinical features. Hoyeraal–Hreidarsson syndrome, a variant of DC, includes reduced intrauterine growth, microcephaly, cerebellar hypoplasia, BM failure and sometimes immunodeficiency with B and NK lymphocyte reduction [181,182]. Other rare inherited disorders, associated with genetically mutated genes involved in telomere protection, are the Revesz syndrome and Coats plus syndrome. These telomeropathies are characterized by intracranial calcification and cysts, exudative retinopathy and gastro-intestinal hemorrhage. Again, BM failure can occur, and it is observed more frequently in Revesz syndrome [183,184].

Some other TBDs are characterized by adult onset of extra-hematological organ failure, including pulmonary fibrosis and emphysema and non-alcoholic steatohepatitis/cirrhosis.

Idiopathic pulmonary fibrosis affects about 20% of DC patients; however, it may occur as a solitary disease manifestation both among familiar and sporadic cases. Disease worsening is usually faster than in patients with pulmonary fibrosis not related to TBD [185]. TBD-related liver diseases include liver cirrhosis, non-alcoholic fatty liver disease, hepatic nodular regenerative hyperplasia, and hepatocellular carcinoma, which may occur in about 7% of all patients with DC. When considering telomere diseases other than DC, liver involvement may occur in a higher percentage of patients. In a cohort of 121 individuals with telomeropathies and variants in *TERT*, *TERC*, and *TINF2*, 40% were found to have some extent of liver disorder [185,186]. Lung and/or liver transplant represent the only definitive therapy for patients with severe lung and/or liver disease, although concomitant cytopenia and multi-organ disease can worsen the outcome [185,187].

Androgen hormones have been demonstrated to activate telomerase and to increase telomere length in telomeropathies [188,189]. In a recent study, nandrolone decanoate was given for two years to 17 patients with short telomeres and/or germline pathogenic variants in telomere biology genes associated with at least one cytopenia and/or radiologic diagnosis of interstitial lung disease. Telomere elongation and cytopenia improvement were observed in 10/16 evaluable patients while improved pulmonary function was observed in 3/7 patients with lung fibrosis. Moreover, pulmonary function worsened again after treatment discontinuation [190].

The individuals who carry a dominant monoallelic variant in a TBD-causing gene can transmit to their progeny not only the deleterious variant but also short telomeres. This can translate into progressive telomere shortening and earlier onset and extent of clinical manifestations over successive generations, a phenomenon known as genetic anticipation. In such cases, the descendants carrying the monoallelic variant exhibit earlier onset of pulmonary disease and other TBD complications, including bone marrow failure [181,191].

#### 3.3.2. Acquired Forms

Telomere has been well-investigated in bone marrow failure disorders arising in subjects with inherited diseases; however, similar attention has been paid to the acquired forms of bone marrow deficiency, namely the acquired Aplastic Anemia (AA) [192]. Since the initial studies, it came out that most patients with AA have markedly shorter telomeres in their PB leukocytes compared to healthy controls, although the degree of TL attrition is variable [193,194]. Telomere attrition is thought to be the consequence of increased proliferative demand on a limited pool of stem cells, following hematopoietic cell destruction operated by the aberrant immune system [163,192]. In other words, at least in most AA patients, short TL is merely a symptom of the underlying disorders, i.e., the abnormal immune-mediated hematopoietic failure. However, it has been then reported that in a minority number of patients with an apparently acquired AA, the pathogenetic mechanism is indeed related to an unrecognized inherited mutation of genes of the telomerase complex, either in TERC or TERT subunits [195,196,197]. This indicates that a minority of patients with supposedly acquired AA harbor inherited telomerase mutations, directly responsible for telomere shortening. Thus, TL shows a wide variation among AA patients, with a great proportion of patients showing markedly short telomeres. Additionally, shortening of TL can be induced by different pathogenetic mechanisms. Lastly, TL of individuals with AA at diagnosis did not significantly differ from that of age-matched subjects. However, short telomeres at diagnosis were linked to outcome, response to immunosuppression, development of cytogenetically aberrant clones and evolution to MDS and AML [198,199,200,201]. Moreover, accelerated telomere attrition precedes progression to monosomy 7 several years before the development of the autonomous and clinically detectable abnormal clone [199,200].

In contrast, patients with AA, who had a treatment response with full hematological recovery, also had their TL returning to normal size compared to age-related controls [193,202]. Remarkably, AA subjects who did not respond to immunosuppressive treatment had considerably short telomeres [203]. Even more relevant is the recent observation that short TL, but not chr7 loss, was associated with worse outcome in patients undergoing allogeneic transplantation for AA [204]. Thus, several reports suggest the value of TL assessment in clinical decisions for patients with SAA.

In summary, the large amounts of studies and reports in both inherited and acquired BM failure syndromes have offered important insights on the role of telomere and telomerase in hematopoiesis, as pathogenetic, diagnostic and prognostic factors for various BM disorders. Moreover, the studies have allowed the improvement in our understanding of the general mechanisms of telomere maintenance and telomere loss. Main studies dealing with telomere and BM disorders leading to BM failure and/or hematopoietic malignancies are summarized in Table 3.

### 3.4. Telomere Length in Hematological Malignancies 

Among hematological malignancies, telomere has been investigated with special interest in myelodysplastic syndromes (MDSs) [210,261,262]. This can be explained by the need of further clues in a malignant disease characterized by an irreversible progression toward acute leukemia with a fatal outcome, although the time to leukemic transformation is often unpredictable. Telomere length showed heterogeneity in MDS, with shorter TL compared to age-matched controls in approximately 50% of patients with MDS. Various reports have shown short TL in peripheral blood leukocytes of MDS while inconsistent results have been reported for non-tumor tissues [213,214,215,216,217,263]. TL shortening at diagnosis correlates with disease severity for several parameters, including the following: i. advanced MDS with excess of blast compared to MDS in early disease phases; ii. complex karyotype abnormalities; iii. adverse features according to the International Prognostic Score System (IPSS) [213,217,218,219]. Thus, telomere shortening can be considered an adverse prognostic marker in MDS indicating a high risk of leukemic transformation and an overall poor outcome [220,221,222,223]. Additional studies suggest a direct link across TL loss, excessive DNA damage accumulation, ineffective myeloid differentiation, dysplasia, and ultimate leukemic transformation [222,224,225].

The natural history of MDS leads to the progressive transformation in acute myeloblastic leukemia (AML). The close link between MDS and AML is reflected by telomere behavior, which is characterized by TL shortening at an analogous degree in AML and in high-risk MDS [226,227,264]. A significantly short TL has been documented in the acute promyelocytic leukemia subtype, although the most pronounced telomere loss has been documented in the acute lymphoblastic leukemia subtypes [228,229,230,232]. Once again, TL has been proven to be of prognostic impact. Indeed, shorter TL has been shown in cytogenetically abnormal leukemic cells compared to leukemic cells with normal karyotype, as well as in cells with multiple aberrations vs. less than two aberrations [226,230,231,265]. Attention has been paid to the activating FLT3-ITD mutation, that is known to be associated with very poor outcome. In fact, the shortest telomeres along with worse survival were seen in AML patients with FLT3 mutations compared to FLT3 wild-type AML [265]. Lastly, patients at relapse showed a shorter TL than at the time of diagnosis [264]. All these observations indicate that TL shortening is quite a common feature in AML and it is closely correlated to chromosome instability and cytogenetic abnormalities.

Besides MDS and AML, neoplasms of the myeloid lineage include a third main group named Chronic Myeloproliferative Neoplasms, which is a group of diseases characterized by bone marrow overproduction of mature blood cells or platelets. There are six types of chronic myeloproliferative neoplasms and Chronic Myeloid Leukemia (CML) is probably the most representative disorder, due to the well-known and widely investigated Philadelphia chromosome, which is an abnormal version of chromosome 22, containing the BCR/ABL, a novel gene made by the fusion of the *ABL* gene and the *BCR* gene. Many studies have found that telomeres of leukemic cells in CML patients are highly variable but generally shorter than those of age-matched healthy individuals or BCR/ABL-negative T lymphocytes from the same patients, along with a slightly increased telomerase activity [233,234,266]. Telomere shortening is accentuated as the disease progresses from the chronic phase to the accelerated and blastic phase and it correlates with the clinical risk score at diagnosis [235,236]. Moreover, TL shortening associates with chromosomal instability, and it has been suggested that telomere shortening in CML might happen before Philadelphia chromosome origination; in other words, non-leukemic cells may already present damage in TL. Therefore, this would contribute to disease onset and progression [237,238,266]. The natural history of CML has dramatically changed since the availability of novel drugs targeting the tyrosine kinase produced by the BCR/ABL fusion gene. Amazing hematological response along with major molecular remission achievement can now be frequently obtained with TK-inhibitors (TKIs) such as imatinib, dasatinib, nilotimib, bosutinib, or ponatinib. Telomere behavior has been investigated as well and it has been observed that the highest responses to TKI can be expected in subjects with long telomeres. Moreover, TL in patients who suffered disease progression exhibited further decrease in telomere length [236,267,268]. On the other hand, a positive correlation between *BCR-ABL* and *hTERT* in a leukemic cell line has been observed, suggesting that suppression of CML cell growth and induction of apoptosis by TKI may be at least partially exerted through telomerase inhibition [269]. All these observations in CML provide further support to the biological and clinical relevance of studies on telomere and TL maintenance system in normal and malignant hematopoiesis.

Other malignant diseases are included in the chronic Myeloproliferative Neoplasia (MPN) group, namely Polycythemia Vera (PV), Essential thrombocythemia (ET) and Primary Myelofibrosis (PMF). These entities are also known as Ph-negative PMNs and are often sustained by mutation of the JAK-2 gene. Studies have been carried out on telomere and telomerase in Ph-negative PMNs and TL ended up being markedly reduced in patients affected by PV, ET, PMF. Consistent with previously reported findings, telomerase activity was upregulated from almost all Ph-neg CMN subgroups compared to healthy donors [239,240,241]. In a collaborative study between Turin and Bergamo groups, TL was found distinctly shortened in PV and MF compared to healthy age-matched subjects and it inversely correlated with JAK2V617F allele burden [242]. The study supports the notion that TL analysis may be exploited as an additional diagnostic marker to identify and monitor Ph-neg-MPN patients. In addition, TL is also of prognostic value in Ph-neg-MPNs, reflecting probably the genetic instability of mutated MPN clones [241,242]. The observation that telomere length can be restored to normal values after treatment with the anti-JAK-2 ruxolitinib gives further support to the role of telomere in the development of MPNs and ultimately its prognostic impact [243]. Previously published data on short telomeres and upregulated telomerase activity in MPNs provided the rationale for novel treatment approaches with the telomerase inhibitor Imetelstat, with the aim of inducing further telomere attrition and inhibiting cell growth in high-risk MPNs [270,271,272]. More studies are required to further characterize the Imetelstat activity with its possible side effects and to identify the exact patient subpopulation who will benefit most from telomerase-targeting treatments [270,273,274]. Nonetheless, the recent results reviewed herein strengthen the importance of ongoing studies on telomeres and telomerase-targeted therapies in myeloid malignancies [275].

Among malignancies of the lymphoid lineage, B-cell chronic lymphocytic leukemia (B-CLL) has probably been the most extensively investigated entity for telomere and telomerase dysfunction over the last 25 years. Since the early studies, it has been clear that a good proportion of B-CLL patients display increased telomerase activity inversely associated with TL reduction. Moreover, both short TL and high telomerase activity were significantly associated with disease progression and reduced survival [244,245]. By that time, it was well-known that peripheral B cell activation takes place within the germinal center (GC), the structures that are transiently formed in secondary lymphoid organs. Accordingly, mature lymphoid neoplasias of the B-cell lineage are distinguished for their origin, either involved or not-involved with the GC activation. The restless cell proliferation within the GC has been substantiated by the observation of high levels of telomerase activity in normal GC B cells in association with a unique telomere lengthening process not observed in other cell types in vivo [246,276,277]. These observations lead our group in Turin to investigate TL in a series of mature B-cell lymphoproliferative disorders classified according to their histopathogenesis in relation to GC. Our study showed that GC-derived tumors have long telomeres, suggesting that minimal telomere erosion occurs during GC-derived lymphomagenesis whereas short TL characterizes GC-inexperienced tumors suggesting that these neoplasms might be good candidates for treatment with telomerase inhibitors [109]. As for the other B-lymphoproliferative disorders, two main subgroups have been identified in B-CLL, with a different clinical course. Some cases have unmutated V(H) genes and are considered to evolve from pre-germinal center (pre-GC) cells. The second main group shows mutated V(H) genes, indicating that the cell of origin is a post-GC derived one. Unmutated pre-GC CLLs are less frequent and have a worse outcome compared to cases of post-GC origin. Telomere status reflects the GC origin. Overall, TL of B-CLL cells is shorter than telomere of B cells from healthy subjects. Moreover, the B-CLL subgroup with limited IgV gene mutations, i.e., pre-GC, have uniformly shorter telomeres and more telomerase activity than those from the IgV gene hyper-mutated subgroup. Thus, differences in TL appear to largely reflect the histopathogenetic histories of precursors of the leukemic cells, and to be associated with a rather different outcome [208,247]. Other groups and we have further investigated the prognostic value of TL in relation to the GC histopathogenesis. It was then shown that TL may identify two different prognostic subgroups among the unfavorable VH-unmutated B-CLL subgroup. Similarly, mutated B-CLLs can be subdivided by TL into two groups with different prognosis. In this way, the combination of TL with V(H) gene mutation status refines prognostication of B-CLL patients, allowing the identification of previously unrecognized prognostic subgroups [209,248].

Subsequent studies, on large and heterogeneous patient populations, confirmed the prognostic value of TL in B-CLL [249,250]. In particular, short TL came out as an independent outcome predictor for both OS and TFS as well as for Richter’s syndrome transformation [249]. Moreover, telomere and telomerase assessments proved to be of useful prognostication for those patients with otherwise good prognostic features. In patients at very early disease onset, a short TL since diagnosis was associated with poor outcome [251,252,253]. Hence, a number of studies and reports have clearly documented the unfavorable prognostic significance of short TL and/or telomerase upregulation in B-CLL. This notion has been consolidated by the observed strong association between short telomeres and the presence of genomic aberrations [254,255]. Short telomeres were remarkably associated with high-risk genetic markers, such as *NOTCH1*, *SF3B1*, or *TP53* aberrations, and understandably predicted an unfavorable outcome [251]. The correlation with *TP53* has been studied with special attention. Any type of *TP53* alteration was found to be associated with very short telomeres and high *hTERT* expression, and patients with disrupted *TP53* showed telomere deletions and chromosomal end-to-end fusions along with complex karyotypes [255,256]. The consistent observations of TL shortening have not only prognostic but also biological implications in B-CLL since telomerase and/or telomeric dysfunctions might be directly involved in the pathogenesis and disease evolution in B-CLL and other malignancies as well. In fact, the recurrent question is whether the short telomeres that characterize most B-CLL cells along with genomic complexity are causative for disease development and clonal evolution or whether telomere attrition is merely the consequence of the complex chromosomal aberrations. This issue is well-summarized in a *Letter to Blood* properly entitled “Short telomeres in B-CLL: the chicken or the egg?” [278].

Genome-wide studies have identified variants linked to human telomere length. These variants can be utilized in Mendelian randomization-based studies to investigate the potential causal relationships between telomere length and cancer development [104]. Based on genotype analysis of large series of patients with lymphoid malignancies and control populations, it has been proposed that subjects with genetic variants associated with long telomeres may be at increased risk of lymphoproliferative disorders, particularly a high risk of B-CLL [205,279]. In other words, long TL might predispose the initiation of B-CLL. A recent study based on Mendelian randomization analysis has confirmed that genetically determined extended telomere length is associated with increased risk of various hematological malignancies, including B-CLL, B-cell lymphoma and multiple myeloma. Moreover, the study highlights the role of epigenetic age acceleration in the development of hematologic malignancies, along with circulating blood cell and biochemical marker abnormalities [206].

In fact, the path linking telomere and B-CLL founding and expansion remains unclear. Some novel clues have been obtained by the demonstration of altered expression of telomerase-binding proteins, shelterin proteins, and multifunctional proteins involved in telomere maintenance in B-CLL cells. The initial studies suggested that telomere shortening already detectable in the early stage of B-CLL might result from specific cell defects in telomerase-dependent and/or telomere-protecting proteins, excluding telomere attrition as a mere consequence of the complex chromosomal aberrations [280,281]. Subsequent exome and Sanger sequencing data from large series of B-CLL patients have allowed the identification of recurrent somatic mutations in *POT1* encoding a component of the telomere-protecting shelterin complex in 3.5% of the cases, and up to 9% of unmutated B-CLL. *POT-1* mutations are responsible for telomere dysfunction and favor the acquisition of the malignant features of CLL cells [282]. This observation indicates how dysfunctional telomeres induce genomic instability and contribute to tumorigenesis. Additional studies will allow the further definition of the genetic feature of telomere dysregulation as a key process in B-CLL development [212].

Among lymphoid malignancies other than B-CLL, telomere and telomerase have been investigated with less interest. Some studies have been reported in the main lymphoma subtypes DLB-CL (Diffuse Large B-cell lymphoma), FL (Follicular Lymphoma) and MCL (Mantle Cell Lymphoma). Basically, telomere length correlates with cellular origin of B-cell malignancies in relation to the germinal center (GC), as shown for the first time by the previously mentioned study of our group [109]. Similarly, to B-CLL, two main DLB-CL subgroups have been identified, those originating from Germinal Center (GC), named GC-like DLBCL, with ongoing mutagenesis and better prognosis and those originating outside the GC, identified as activated B cell-like (ABC) DLBCL, with poor prognosis [283]. Overall, telomere length is heterogeneous in DLB-CL, and GC-like DLBCL originating from GC has longer telomeres than post-GC ABC-DLBCL, with a subset of the GC-like DLBCL accounting for the longest TL [109,210,211]. In FL, TL is largely similar to DLBCL, although shorter than GC-DLBCL. Compared to DLB-CL and FL, MCL cells show shorter TL; however, their TL is longer than in IGHV-unmutated CLL cells and shorter than IGHV-mutated CLL cells. Thus, TL is not simply determined by GC-related origin; it may reflect other parameters which have not been clarified yet. In line with this hypothesis, TL has been found to be linked with the presence of BCL2 gene rearrangements resulted from chromosomal t (14;18) translocation, as shown in a series of patients with FL and a small group of aggressive lymphoma transformed from FL. Overall, FL patients with bcl-2 gene rearrangement had longer TL than those without. The shortest TL was found in aggressive forms transformed from an FL [207]. With regard to lymphoma development and telomeres and/or telomerase dysfunction, a study analyzed mononuclear cell DNA of pre-diagnostic peripheral blood samples from 464 lymphoma cases and 464 matched controls, with a median time between blood collection and lymphoma diagnosis of 4.6 years. TL was significantly longer in cases developing lymphoma compared to age- and gender-matched controls. The study confirms an association between longer leucocyte TL and increased risk of B-cell lymphoid malignancies, which appeared most pronounced for DLBCL and FL [284]. Other genetic studies have proposed this viewpoint relating longer telomere length with increased lymphoma risk [205]. Nevertheless, short TL remains independently associated with poor outcome following standard chemo-immunotherapy in B-cell lymphoid malignancies [285]. The well-documented TL abnormalities have promoted further studies on telomere, telomerase, and associated proteins in lymphoma. A recent report has found a significantly lower expression of some genes belonging to the telomere-related genes (TRGs) group in DLBCL lymph node samples compared to normal tissues. Two distinct subgroups have been molecularly identified based on expression patterns of telomere-related genes. Differences in gene expression allowed the development of a novel prognostic model in DLBCL highly predicting for treatment responsiveness and overall outcome [286]. This confirms how ongoing studies may improve our insights on the role of telomere dysfunction in lymphoid malignancies.

The vast group of B-lineage malignancies includes one more main disorder, the Multiple Myeloma (MM), which is recognized as the abnormal proliferation of malignant plasma cells. These latter cells represent the final differentiation step of activated peripheral B-cells. MM may arise de novo or it can evolve from a pre-malignant condition known as monoclonal gammopathy of undetermined significance (MGUS). Similarly to the other malignant disorders, TL along with telomerase activity, has been investigated with interest in both MM and MGUS, and the results are quite comparable to what has been previously described. Indeed, since the initial observations performed on purified CD138+ plasma cells, TL was found to be significantly shorter than that of the patient’s own leukocytes in a good proportion of cases. Telomerase activity was heterogeneous, with high levels in a subgroup of patients. Indeed, telomerase activity was correlated to TL, suggesting a telomerase-mediated stabilization of short telomeres when TL fell below a critical threshold [257]. A concomitant study confirmed in a group of MM a mean TL value shorter than that observed in controls. However, this was not seen in MGUS subjects. Additionally, a significant TL shortening was observed at disease relapse compared to diagnosis, whereas TL was restored in case of remission achievement [258]. Both studies documented that cytogenetic abnormalities, including those associated with poor prognosis, correlated with TL shortening and telomerase activation, supporting the view of short TL accountable for increased chromosome instability and associated with clinical evolution. Thus, TL and telomerase activity were again proposed as useful prognostic factors for MM [257,258,259]. A subsequent study in a large patient cohort showed that MM patients with short telomeres had a significantly reduced overall survival, suggesting the inclusion of a TL parameter as a refinement of the International Staging System (ISS) [260]. A more recent analysis on 251 MM patients has clearly shown that TL was significantly shorter in MM than in healthy controls, and patients with more advanced disease according to ISS had shorter TL than patients with less advanced disease. The same study has also identified specific genetic variants of telomerase that were found to be less common in patients with poor treatment response [287]. In order to characterize the mechanisms of telomere dysfunction that may contribute to MM expansion, telomere-related genes as well as telomerase gene expression have been thoroughly investigated and found altered, similarly to previously mentioned reports in CLL and other lymphoid malignancies [288,289,290,291]. The gene signature studies indicate some possible explanations for the maintenance of short TL along with tumor expansion in MM. All these observations can be exploited to design novel treatment strategies based on proteasome inhibitors or other effective drugs for MM, given that their reported activity targeted telomerase and/or telomere-associated proteins [292,293,294].

To sum up, telomere has been extensively investigated in BM failure disorders and in hematological malignancies. The occurrence of TL shortening is quite a common feature in these conditions, and it is often associated with major genomic rearrangements and DNA damage. The studies conducted so far and reported in this chapter have markedly enhanced our understanding on telomere, telomerase, and associated proteins, allowing a better prognosis of patients in the clinical practice and paving the way for future therapeutic applications. The vast literature on telomere and hematological neoplasms is summarized in Table 3, as previously mentioned.

## 4. Summary and Conclusions

Over the past three decades, numerous studies have focused on chromosome telomeres and their role in regulating cell senescence, the pathogenesis of degenerative diseases, and carcinogenesis. Telomere shortening is a common consequence of successive cell divisions, limiting the proliferative lifespan of cycling cells by inducing quiescence or apoptosis once a critical telomere length has been reached. However, several regulatory mechanisms modulate the telomere response to cell proliferation in different tissues. In particular, during embryogenesis and in adult stem cells, telomerase and other enzymes act to restore telomere length after each cell division. A similar mechanism generally occurs in tumor cells at a certain stage of carcinogenesis.

Telomere length assessment has been recognized as a valuable tool for understanding the mechanisms of cellular aging and cancer development, and several methods have been proposed to measure telomere length. Each technique offers specific advantages, but none is completely free of limitations or universally applicable. Therefore, the choice of the most appropriate method for telomere measurement should depend on the study design and the available cell source.

Current evidence has demonstrated a role of defective telomere re-elongation in several hematological disorders, such as bone marrow hypoplasia, which may result either from genetic defects in telomere repair or from autoimmune damages to hematopoietic stem cells, leading to proliferative stress on the residual stem cell population. In immune-related aplastic anemia, telomere re-elongation is often observed in patients responding to immunosuppressive therapies. It remains unclear whether this is due to the selection of a small population of stem cells with longer telomeres or to marked telomerase reactivation.

The role of telomere length in carcinogenesis is more complex and not yet fully understood. Telomere shortening, by limiting the number of cell divisions, is generally regarded as a protective mechanism against tumorigenesis. Indeed, some individuals with genetically determined long telomeres appear to have an increased risk of certain malignancies, such as gliomas and early-stage hematological neoplasms, including CLL and some non-Hodgkin’s lymphomas. Conversely, excessively short telomeres increase the likelihood of chromosomal fusions and other genetic abnormalities. Fully malignant cells, however, typically evade telomere-induced growth arrest through mutations in genes involved in the DNA damage response, most notably *TP53*, and by preventing further critical telomere shortening via telomerase reactivation or homologous DNA recombination mechanisms.

In fact, most malignant tumors, including colon–rectal, lung, breast, and pancreatic cancers, as well as acute leukemias, high-risk myelodysplastic syndromes, myeloproliferative neoplasms, and certain lymphoproliferative disorders, are characterized by short telomeres despite increased telomerase activity. In many of these cases, telomere shortening correlates with the presence of genetic abnormalities associated with poor prognosis. Thus, it appears evident that both excessively short and abnormally long telomeres can predispose one to tumorigenesis.

Nevertheless, many questions remain unresolved. These include defining the most reliable and standardized method for measuring telomere length; explaining the apparent paradox of short telomeres coexisting with increased telomerase activity in neoplastic cells; identifying potential drugs or procedures capable of preventing excessive telomere shortening with aging; and, conversely, determining which neoplasm may benefit from telomerase inhibitor therapies. Further research is required to address these crucial issues within the fascinating field of telomere biology, cellular aging, and cancer development.

## Figures and Tables

**Figure 1 biomedicines-13-03009-f001:**
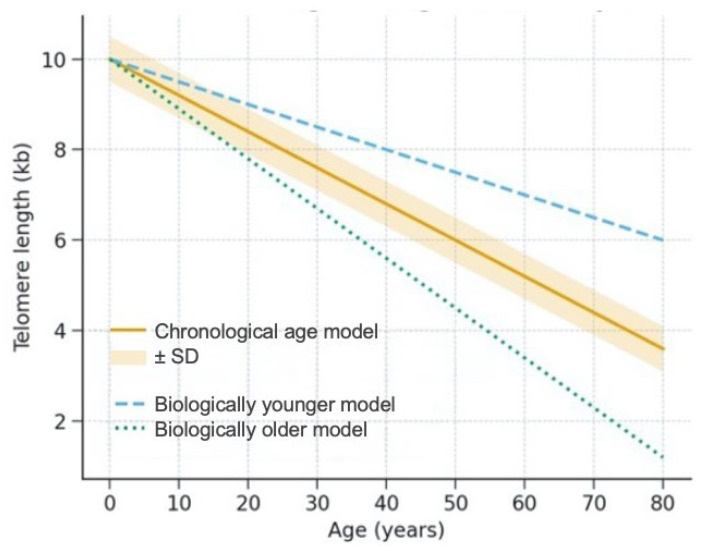
Telomere length dynamics according to chronological age and biological aging. Telomere length, measured in kilobases (kb), is plotted against chronological age (years), showing the canonical age-dependent telomere attrition. The central regression line (*Chronological age model*) represents the expected trajectory in the general population as reported in the literature (usual loss of approximately 60–70 bp/year), with shaded bands indicating the standard deviation around the mean (≈±0.5 kb), consistently with values commonly reported in large-scale cohort studies. A conceptual model of telomere trajectories has been conceived showing the following: *i.* a *biological younger model*, showing a slow telomere shortening associated with favorable biological and environmental influences (e.g., intact telomerase activity, low oxidative stress, healthy lifestyle factors); *ii.* a *biological older model*, characterized by enhanced telomere loss, due to chronic exposure to a number of exogenous factors adversely influencing telomere length maintenance. Individuals displaying biological age acceleration fall below the expected telomere length for their chronological age, whereas those with slower biological aging maintain longer telomeres relative to age-matched peers. In some cases, genetic factors can predispose one to longer telomere length, alternatively, germline mutations are known to cause progressive and abnormal telomere loss. Image editing was assisted by ChatGPT-5.1, 2024 [36] and broadly adapted by the authors.

**Figure 2 biomedicines-13-03009-f002:**
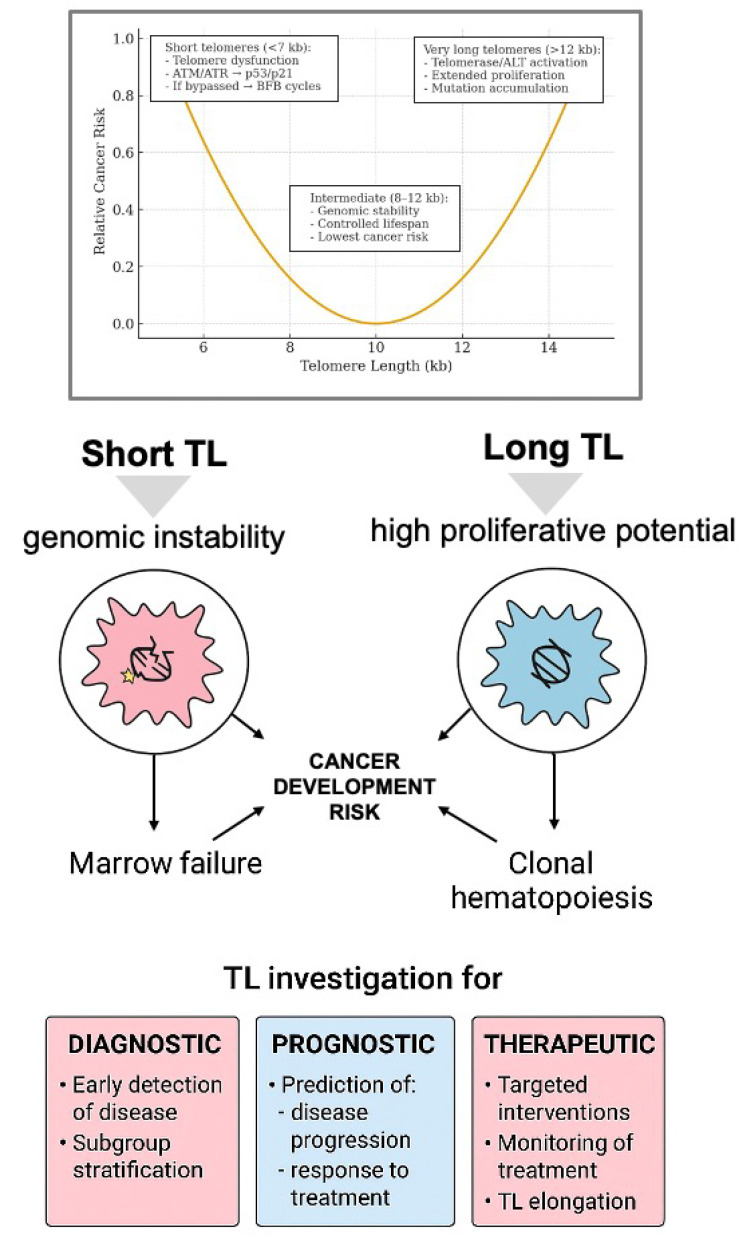
Role of telomere in cancer development and the value of telomere length assessment. Conceptual model illustrating the non-linear relationship between telomere length (TL) and cancer risk. Extremely short TL (<~7 kb) triggers the complex DNA damage response (DDR), involving the ATM/ATR pathway leading to cell arrest and apoptosis mediated by activated p53 and p21. If the DDR process is severely compromised, short TL drives genomic instability, with Breakage-Fusion-Bridge (BFB) cycles, that favor the initiation and progression of cancer cells. On the other hand, excessively long TL (>~12 kb) supports extended proliferative potential and clonal evolution, again increasing malignant transformation risk. Intermediate TL (8–12 kb) maintains controlled proliferation and genomic stability, associated with the lowest cancer risk. Below, a schematic representation highlights the clinical consequences of short and long TL: basically, dysfunctional TL may lead to increased cancer risk; in addition, short TL predisposes one to bone marrow failure, while long TL favors clonal hematopoiesis. Both these features may ultimately end up in cancer development. Due to the marked involvement in carcinogenesis, TL assessment has diagnostic value (early disease detection and patient stratification), prognostic utility (prediction of disease progression and treatment response), and therapeutic relevance (monitoring of telomere-targeted interventions and potential TL elongation strategies). Image editing was assisted by ChatGPT [36] and broadly adapted by the authors.

**Table 1 biomedicines-13-03009-t001:** Synopsis of techniques available for telomere length analysis.

Type of Procedure ^1^	Product	DNA Amount or Number of Cells Required	Time Needed for the Assay (h)	Distinct Feature	Reference
**TRF**	average/absolute TL	>1 µgr	>48 h	gold standard	[11]
**Q-PCR**	average/relative TL	20 ng	<2 h	high throughput	[64]
**Q-FISH**	relative TL	10–15 cells	>72 h	feasible on formalin-fixed paraffin-embedded tissue	[65]
**Flow-FISH**	average/relative TL	1 × 10^5^ cells	>72 h	well-suited for combination with additional flow cytometry analysis	[66]
**TIF**	DNA damage	>10–15 cells	24–48 h	developed for telomere dysfunction detection	[67]
**STELA**	absolute TL	10–50 ng	>72 h	particularly suitable for measuring shortest TL	[68]
**TeSLA**	absolute TL	10–50 ng	>72 h	high accuracy and sensitivity	[69]
**SCT-pqPCR**	average TL	10 pg	>2 h	developed for TL analysis in single cells	[70]
**Optical Mapping**	absolute TL	20 µgr	>24 h	developed for TL measurement in each chromosome	[71]

**^1^** Designation: TRF, Telomere Restriction Fragment analysis by Southern blotting; Q-PCR, Quantitative Polymerase Chain Reaction; Q-FISH, Quantitative Fluorescence In Situ Hybridization; Flow-FISH, flow cytometry fluorescence in situ hybridization; TIF, telomere dysfunction-induced foci analysis; STELA: Single Telomere Length Analysis; TeSLA, Telomere Shortest Length Assay; SCT-pqPCR: single-cell telomere length measurement by pre-amplification and quantitative PCR; Optical Mapping, single-molecule telomere length characterization by optical mapping; TL, Telomere Length.

**Table 2 biomedicines-13-03009-t002:** Summary of telomere length deviations in non-hematologic cancers.

Telomere Length ^1^	Type of Neoplasm	Reference
**Longer than in normal subjects: germ line genetic variants**	▪ Increased risk (OR 2.98–5.27) of the following:	[130]
	Glioma/glioblastoma	
	Lung cancer (mainly adenocarcinoma)	
	Serous ovarian cancer	
	Neuroblastoma	
	▪ Thyroid (OR 2.49) and lung (OR 2.19) cancers	[131]
**Shorter than in normal subjects and/or normal tissues, with increased telomerase activity**	▪ Analysis on 18,430 samples of 31 cancer types: 70% had short telomeres	[54]
	▪ Colon–rectal cancers (particularly in Lynch syndrome)	[137]
	▪ Colorectal cancers	[138,148]
	▪ Pre-neoplastic pancreatic lesions (increased risk of neoplastic evolution)	[139,140,141]
	▪ Pancreatic ductal adenocarcinomas	[142]
	▪ Early lung, breast and prostate cancers	[143]
	▪ Breast cancers	[144]
	▪ Gastric cancers	[148,149,150]
	▪ Barrett esophagus (with increased risk of cancer evolution)	[146]
	▪ Esophageal adenocarcinoma	[147]
	*Short telomeres generally associated with poor prognosis neoplasms*	

**^1^** Telomere length determined either on PB leukocytes or on both malignant and normal tissues. Main methods used for telomere length evaluation in the quoted references (ref.): Southern blotting (ref. [148]); Quantitative Fluorescence In Situ Hybridization (Q-FISH) (ref. [139,141,142]); multiplex real-time PCR (ref. [138]); nested real-time quantitative PCR (ref. [140]); real-time quantitative PCR (ref. [146,150]); whole-genome sequencing (ref. [54,137]). Genome-wide association studies (GWAS) (ref. [130,131]). Not specified, or TL not determined, or telomerase/other genes evaluated (ref. [143,144,147,149]).

**Table 3 biomedicines-13-03009-t003:** Summary of telomere length deviations in hematological malignancies and other disorders predisposing one to hematological malignant transformation.

Telomere Length ^1^	Type of Neoplasm	Reference
**Longer than in normal subjects: germ line genetic variants**	▪ Clonal Hematopoiesis of Indetermined Potential (CHIP)	[103,107]
	▪ Increased risk of B-CLL, B-cell Lymphoma, multiple myeloma	[205,206,207]
**Longer or equal than in normal subjects, i.e.: B cell neoplasms originating from lymphoid germinal centers**	▪ Chronic lymphocytic leukemia arising from germinal center B lymphocytes: V(H)mutated	[208,209]
	▪ Diffuse Large B-cell Lymphomas originating from germinal center	[109,210,211]
	▪ Follicular lymphomas (initial stages) with BCL2 rearrangement	[207]
**Short due to germline-inherited conditions known as telomere biology disorders (TBDs)**	▪ Inherited telomeropathies including	
	i. Dyskeratosis Congenita (DC)	[157,158,159,160,161,162,163]
	ii. Inherited BM failure (IBMF)	[174,175]
	iii. Inherited form of Aplastic Anemia	[163,195,196,197]
	iv. Clonal Hematopoiesis of Indetermined Potential (CHIP)	[167,169]
	v. Familial hematopoietic malignancies	[164,169,212]
**Shorter than in normal subjects**	▪ Acquired Aplastic Anemia	[192,193,194,198,199,200,201]
**Shorter than in normal subjects and/or normal tissues, with increased telomerase activity**	▪ High-risk myelodysplastic syndrome	[213,214,215,216,217,218,219,220,221,222,223,224,225]
	▪ Acute myeloid leukemias (particularly with unfavorable cytogenetic or FLT3-Itd)	[226,227,228,229,230,231]
	▪ Acute lymphoblastic leukemia	[230,232]
	▪ Chronic myeloid leukemia (particularly in progression)	[233,234,235,236,237,238]
	▪ Polycythemia Vera, Essential Thrombocythemia, Primary Myelofibrosis	[239,240,241,242,243]
	▪ Chronic lymphocytic leukemia V(H) unmutated (not originating from germinal center), particularly in progression	[208,209,244,245,246,247,248,249,250,251,252,253,254,255,256]
	▪ Diffuse Large B Cell Lymphoma (DLBCL) non originating from germinal center	[207]
	▪ Follicular Lymphoma in transformation	[207]
	▪ Multiple myeloma with unfavorable cytogenetic.	[257,258,259,260]

**^1^** Telomere length determined either on PB leukocytes or on both malignant and normal tissues. Main methods used for telomere length evaluation in the quoted references (ref): Southern blotting (ref. [109,193,207,208,213,217,218,219,220,228,229,230,235,239,242,244,257,258]); quantitative or flow-cytometry-based fluorescence in situ hybridization (Q-FISH, flow-FISH) (ref. [107,157,161,194,195,196,197,212,222,225,226,231,232,234,236,237,247]); modified T/C FISH (ref. [215,221,241]); real-time quantitative PCR (ref. [198,199,200,211,223,240,243,250,251,252,254]); use of various methods (Q-PCR and/or Southern blotting and/or Q or flow-FISH and/or others) (ref. [162,164,209,214,216,238,255,256]); Single Telomere Length Analysis (STELA) (ref. [227,253,260]); not specified, or TL not determined, or telomerase/other genes evaluated (ref. [103,158,159,160,167,169,174,175,192,201,205,206,210,224,233,245,246,259]).

## Data Availability

No new data were created or analyzed in this study.

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
