# Peer review of "Telomere Length Abnormality: Investigating Approaches and Correlations with Cancer, Bone Marrow Failure and Hematological Malignancies"

_biomedicines, 2025, doi:10.3390/biomedicines13123009_

Round 1

Reviewer 1 Report

Comments and Suggestions for Authors

This review provides a comprehensive overview of telomere biology, measurement methods, and telomere length in disease. Overall, the manuscript is well-written, most terms are introduced with concise and accurate background information, and the inclusion of recent therapeutic advances is valuable. Nevertheless, I have several suggestions for improvement:

  1. Index: qPCR appears twice (II and IV). Please revise.
  2. Introduction: Language could be more concise and precise.
  3. Nomenclature: Please use consistent terminology (TERT hTERT) and follow HGNC guidelines to distinguish gene symbols (italicized) from protein names.
  4. Section 3.2.1: “In Main Cancer Types” may be better phrased as “In Solid Tumors.” Hematological malignancies also represent a major class of cancers. When I finished this section without finding them, it felt incomplete, as telomere length is an unavoidable issue in blood cancers.
  5. Telomere biology disorders: Given the increasing understanding of telomere biology disorders, a dedicated section is warranted. Accumulating evidence indicates that certain patients with pulmonary fibrosis or other fibrotic diseases (without concurrent bone marrow failure) in fact have TBDs. The review should summarize these findings and highlight the unique hereditary nature of telomere-related gene germline mutations, as well as their role in genetic anticipation (earlier onset in successive generations).

Reviewer 2 Report

Comments and Suggestions for Authors

The manuscript by Torella et al., “Telomere Length Abnormality: Investigating Approaches and Correlations with Cancer, Bone Marrow Failure and Hematological Malignancies,” reviews recent advances in the telomere length biology field. Telomere length (TL) is a critical parameter implicated in numerous biological processes and associated with various diseases, including aging and cancer. The authors provided an extensive overview of the methods for TL detection and discussed recent studies on the association between TL and cancer and hematological malignancies. The authors review a significant question in their manuscript. However, there are a few issues that should be addressed. Here are my comments.

Index. It is helpful to have an index at the beginning of the paper. However, the index listing does not correspond to the text. Also, Numbering in the text is different. The QPCR technique is listed twice.

Figure 1. Description in the text is too general and does not correspond to the details in the figure “clonal hematopoiesis”, “marrow failure”. It would be beneficial to add a more detailed description of the text or figure legends.

Authors describe in detail the implications of TL in specific diseases; however, expanding the chapter on the general concept of telomere length regulation would be helpful for the reader.

Reviewer 3 Report

Comments and Suggestions for Authors

In the manuscript “Telomere Length Abnormality: Investigating Approaches and Correlations with Cancer, Bone Marrow Failure, and Hematological Malignancies,” the authors have performed a broad and contemporary review of telomere length abnormalities and their clinical correlations, particularly focusing on cancer, bone marrow failure, and hematological malignancies. The article covers current telomere length measurement techniques and summarizes associations between telomere length and disease. The subject is interesting, but some issues should be addressed.

1-The legend accompanying Figure 1 lacks sufficient detail, and the figure itself is not highly informative in its current form. I recommend adding more explanatory content to the legend.

2-In Table 1, the unit “mcgr” is used for DNA quantity. To ensure clarity and consistency, this should be corrected to the standard scientific notation “μg” (micrograms).

3- Consider incorporating additional illustrative models or diagrams to aid rapid comprehension of key concepts, particularly for complex biological processes.

4- For Table 2, it would be helpful to specify which telomere length measurement method was used in each cited study.

5-The relationship between patient age and sex and telomere length deserves further discussion, especially when interpreting telomere length in patients.

Round 2

Reviewer 3 Report

Comments and Suggestions for Authors

All comments have been addressed, and the manuscript is now ready for publication.